# A scoping review of interventions to prevent and treat adverse events during treatment of rifampin-susceptible tuberculosis

William J. Burman[1,2]*, M. Florencia Martins[3¤], David Flynn[4], James Johnston[5,6], Pranay Sinha[3], C. Robert Horsburgh[7]

1 Public Health Institute at Denver Health, Denver, Colorado, United States of America, 2 Department of Medicine, University of Colorado Anschutz Medical Campus, Aurora, Colorado, United States of America, 3 Section of Infectious Diseases, Boston Medical Center, Boston, Massachusetts, United States of America, 4 Department of Medical Sciences and Education, Boston University Chobanian & Avedisian School of Medicine, Boston, Massachusetts, United States of America, 5 Provincial TB Services, British Columbia Centre for Disease Control, Vancouver, British Columbia, Canada, 6 Faculty of Medicine, University of British Columbia, Vancouver, British Columbia, Canada, 7 Departments of Global Health, Epidemiology, Biostatistics and Medicine, Schools of Public Health and Medicine, Boston University, Boston, Massachusetts, United States of America

¤ Section of Infectious Diseases, Beth Israel Deaconess Medical Center, Boston, Massachusetts, United States of America
* bill.burman@dhha.org

## Abstract

### Background

Treatment-related adverse events are one of the leading barriers to tuberculosis treatment completion but have not been the focus of late-phase clinical trials. We performed a scoping review to identify interventions to improve the safety and tolerability of rifampin-susceptible tuberculosis. Our objective was to determine what interventions have been evaluated to prevent or manage adverse events, as well as what research is underway.

### Methods and findings

We searched Embase, PubMed, Cochrane Central Register of Controlled Trials, Cochrane Database of Systematic Reviews, and Web of Science from 1970 to December 2024 using a broad set of terms regarding adverse events, as well as citation searches to identify additional studies in topic areas that were not well-represented in the initial title search. To identify research in progress we searched Clintrials.gov, Cochrane reviews, and International Clinical Trials Registry Platform for trials reported to be active between January 2015 to April 2025. Of 7314 titles reviewed, 119 papers were available and eligible for this scoping review: 37 (31%) evaluated changes in the tuberculosis treatment regimen, 55 (46%) evaluated other interventions to prevent adverse events, and 27 (23%) evaluated treatment of

**Data availability statement:** All relevant data are within the Supporting Information file.

**Funding:** The author(s) received no specific funding for this work.

**Competing interests:** The authors have declared that no competing interests exist.

adverse events. Only 7 studies reported enrollment of children < 12 years old. Of the 49 clinical trials, 20 (41%) had sample sizes < 50 participants/arm. Notable gaps in research in this field: uncertainty about the safety of pyrazinamide, lack of research on prevention and management of nausea/vomiting, uncertainty about the impact of hepatoprotectants, and lack of inclusion of children. Of the 8 study proposals that appear to be in progress, five were for a single topic: isoniazid dosing based on N-actyltransferase-2 status.

## Conclusions

There has been considerable research on improving the safety and tolerability of tuberculosis treatment, but its impact is limited by under-powered studies, the lack of inclusion of key subgroups, and important gaps in the research portfolio (uncertainties about the safety of pyrazinamide and the efficacy of hepatoprotectants, lack of research on ways to manage and prevent treatment-related nausea). It is concerning that the research pipeline for interventions to improve safety and tolerability appears to be quite limited Our review has identified promising interventions that may make treatment better tolerated, and hence, more effective.

## Introduction

Clinical trials and cohort studies suggest that one of the leading barriers in curing rifampin-susceptible tuberculosis is the occurrence of treatment-related adverse events. Adverse events are common [1–4], and have been associated with missed doses and treatment interruptions [5–7], regimen changes and extensions [8,9], treatment non-completion [10], and increases in treatment failure, recurrence, or death [1,4,11]. Moreover, some populations, such as the elderly and persons with HIV co-infection [12], diabetes [13], and alcohol use disorder [14], are at higher risk for drug intolerance and adverse events, resulting in worse outcomes in programmatic settings [8,9]. Common treatment-related adverse events include nausea/vomiting, hepatotoxicity, skin rash, peripheral neuropathy, visual disturbances, drug fever, and arthralgias [2].

Despite the frequency of adverse events and their impacts on achieving tuberculosis cure, the focus of clinical trials over the past 20 years has been on treatment-shortening, rather than improving safety and tolerability of therapy. Only 5 of 40 late-phase clinical trials published between 2000–2023 evaluated interventions for reducing adverse events during the treatment of rifampin-susceptible tuberculosis [15]. Improving the outcomes of tuberculosis treatment will require interventions that prevent or better manage common adverse events, as well as the identification of new regimens with better tolerability [16]. The purpose of this scoping review is to provide an overview of published and planned research on approaches to managing treatment-related adverse events that can be the basis for developing a research agenda for future studies of interventions to improve the safety and tolerability of tuberculosis treatment.

## Methods

This scoping review was done using the JBI guidelines [17] and the checklist for scoping reviews of the report for systematic reviews and meta-analyses (PRISMA-ScR) [18] (S1 Table). We developed four guiding questions for this scoping review:

- What interventions have been evaluated to prevent or manage common treatment-related adverse events during treatment for rifampin-susceptible tuberculosis (nausea, hepatoxicity, hypersensitivity, retinal toxicity, neuropathy, arthralgia)?

- What research is underway in this field?

- What sample sizes have been used in studies in this field?

- What are key gaps in the literature regarding minimizing and managing adverse events?

We then developed a framework for categorizing possible interventions to improve safety and tolerability of tuberculosis treatment (Table 1).

## Search methods

We searched Embase, PubMed, Cochrane Central Register of Controlled Trials, Cochrane Database of Systematic Reviews, and Web of Science from January 1970 to December 2024 using a broad set of terms regarding adverse events (S2 Text). We searched ProQuest Dissertations and Theses Global database for PhD theses from 2000 to 2025. Following an initial title review, we mapped the results to the framework in Table 1 to identify topics that had not yielded many titles. We identified seed articles on under-represented topics from the initial search and from articles in the authors' files. We then used these seed articles to do forward and backward citation searches [19].

Finally, we searched Clintrials.gov, Cochrane reviews, and International Clinical Trials Registry Platform for trials listed as being active between January 2015 to April 2025 using the search terms "tuberculosis and side effects" and "tuberculosis and toxicity" to identify unpublished trials evaluating interventions to improve safety and tolerability.

**Table 1. Framework for categorizing possible interventions to address treatment-related adverse events.**

| |
|---|
| ***Prevention of adverse events*** |
| Tuberculosis treatment regimen changes |
| • Pyrazinamide – dose, duration |
| • Isoniazid – dose, alternative drugs |
| • Rifamycin – choice, dose |
| • Ethambutol – duration, alternative drugs |
| • Dosing frequency – split dose, intermittent dosing |
| • Duration of therapy |
| Other interventions to prevent specific adverse events |
| • Nausea/vomiting |
| • Hepatotoxicity |
| • Immunomodulatory agents |
| ***Treatment of adverse events*** |
| • Nausea/vomiting |
| • Hepatotoxicity |
| • Immune hyper-reactivity |
| • Hypersensitivity reactions |

## Inclusion and exclusion criteria

We included experimental and quasi-experimental study designs including randomized controlled trials, prospective and retrospective cohort studies, case-control studies, and systematic reviews. Studies of adults and children were included. Papers must have included an evaluation of one or more interventions to decrease the risk of adverse events or to treat adverse events occurring during treatment of active tuberculosis. Our primary focus is on the treatment of rifampin-susceptible tuberculosis, but we included studies on the treatment of rifampin-resistant tuberculosis if they addressed interventions for the common adverse events listed above.

We excluded studies on tuberculosis preventive treatment and studies focusing on interventions regarding drugs that are not currently recommended for treatment of rifampin-susceptible tuberculosis (e.g., thiacetazone, aminoglycosides, cycloserine, bedaquiline, delamanid, pretomanid, clofazimine). We also excluded studies on the epidemiology of adverse events, including pharmacogenomic studies, if they did not include an evaluation of an intervention to decrease adverse events. We did not review the individual studies included within the systematic reviews that were included in the scoping review.

Finally, we did not formally evaluate study quality

## Selection of articles

Article titles were downloaded into Covidence, deduplicated, and reviewed by two of the authors (WB, MFM, PS). Articles in languages other than English or Spanish were translated using ChatGPT.

Full text reviews and data abstraction of the papers that met the inclusion/exclusion criteria were performed by WB and MFM. Differences between the two reviewers were resolved by consensus. The results of the search of clinical trials registries were reviewed by CRH and PS and checked against the papers in the scoping review to identify trials that had subsequently been published. The full team met monthly to review progress and make decisions about methodological issues (e.g., need for citation searches, ways to handle multiple systematic reviews of a specific topics). The protocol was registered at Open Science Framework on 21 April 2025.

## Data extraction and analysis

Data abstraction was done using an Excel form including study type, country in which the research took place, intervention evaluated, number of study participants (average number/arm in randomized clinical trials), inclusion of persons < 18 years of age, and a brief summary of the results. We defined trials having < 50 participants per arm as being small and likely to be under-powered. The full team then reviewed the evidence available in each of the topic areas to identify high-priority research needs. All analyses were descriptive only.

## Results

Of 7314 unduplicated titles identified, 217 papers were selected for full-text review (Fig 1). Twenty papers could not be retrieved; these papers focused primarily on non-standardized hepatoprotectants and other forms of prevention (S3 Table). Of the remaining 197 papers, 78 were ineligible on full-text review (Fig 1). The 119 papers included were from around the globe, most frequently from India [20], China [15], Japan [9], South Africa [8], and Iran [7]. The study methodology and sample sizes of the papers are summarized in Table 2.

Of the 119 papers, 37 (31%) evaluated changes in the treatment regimen, 55 (46%) evaluated interventions other than changes in the treatment regimen to prevent adverse events, and 27 (23%) evaluated treatment of adverse events (S4 Table). Ten papers in languages other than English or Spanish were translated using ChatGPT (S4 Table). Of the 49 randomized controlled trials, 20 (41%) had < 50 participants per arm, hence meeting our definition as likely to be under-powered (S5 Table).

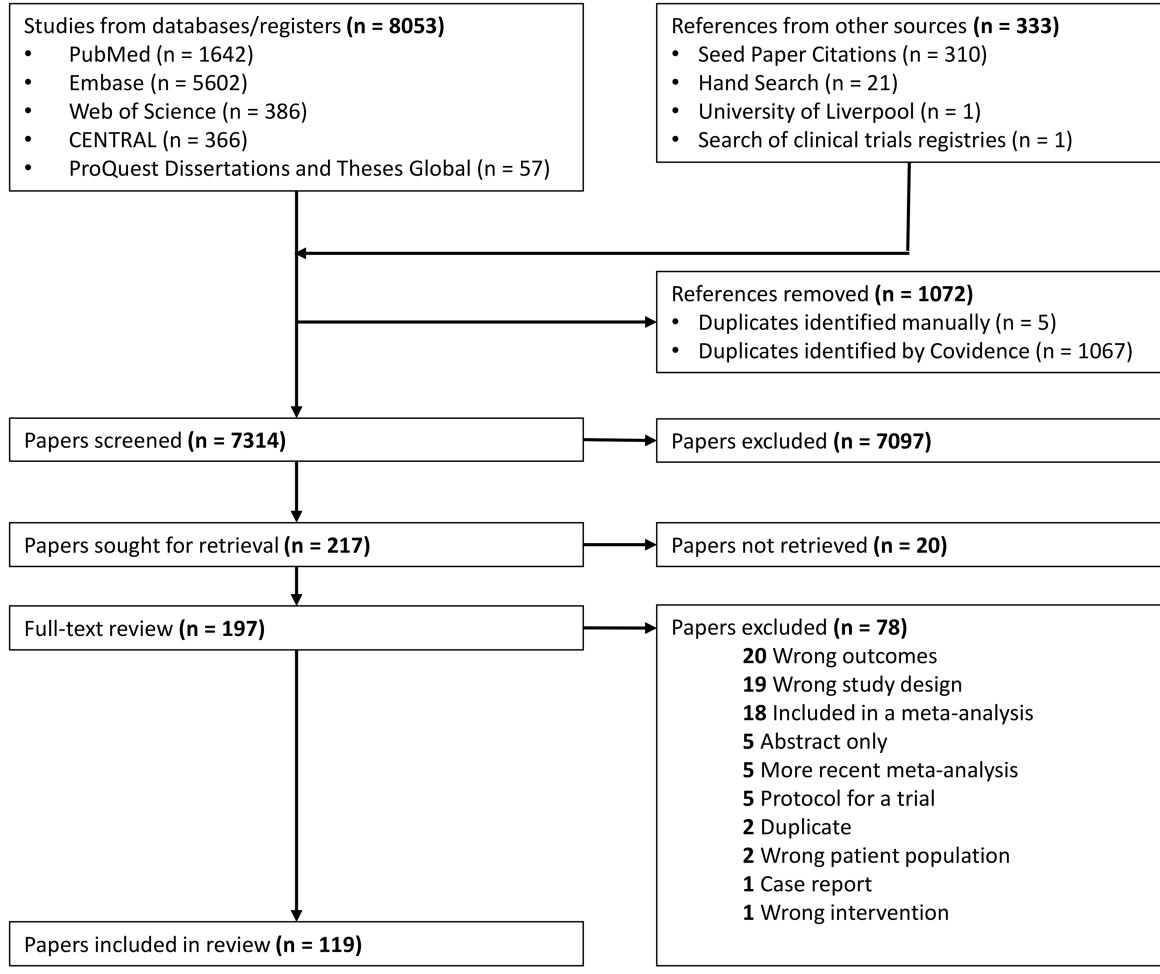

**Fig 1. PRISMA flow diagram of study selection.**

**Table 2. Characteristics of the studies included.**

| Study methodology | N | Median sample size, IQR (range) |
|---|---|---|
| Retrospective cohort studies | 22 | 103.5, IQR 46–488 (range 4−7,156) |
| Prospective cohort studies | 24 | 67.5, IQR 21–198 (range 6−4,488) |
| Randomized controlled trials | 49 | 60/arm, IQR 30–120 (range 5−781) |
| Systematic review/meta-analysis | 24 | 2,540, IQR 686–5,824 (range 64−15,586) |

We evaluated the inclusion of persons < 18 years of age. Excluding systematic reviews (which frequently did not report whether persons < 18 years were included), 65 of the 95 remaining studies (68%) were limited to persons ≥ 18 years, 12 studies (13%) enrolled adolescents (12–17 years), 7 studies (7%) reported enrollment of children (< 12 years), and 11 studies (12%) did not report this information (S6 Table).

Of the 538 entries in clinical trials registries, 17 study proposals were for eligible studies focused on improving safety and tolerability, 9 of which matched to papers in the scoping review (S7 Fig). Of the 8 remaining study proposals

(S8 Table), five were for the evaluation of customized isoniazid dosing using N-acetyltransferase 2 enzyme (NAT-2) activity, and three were evaluations of putative hepatoprotectants (Vitamin C, N-acetyl cysteine).

## Changes in the TB treatment regimen

More than a third (14/37) of papers on changes in the treatment regimen focused on the use, dose, and duration of pyrazinamide (Table 3). The results of these studies are mixed, exemplified by two systematic reviews of the safety of pyrazinamide, one of which concluded that neither its inclusion nor dose in treatment regimens affected rates of hepatoxicity [20] and the other which found an increase in adverse events and regimen changes (though not hepatoxicity) [21]. A recent small, randomized trial of persons 80 and older found no increase in hepatotoxicity but an increase in death among those randomized to pyrazinamide [22]. However, three cohort studies did not find increased hepatoxicity or death among persons ≥ 65 years who received pyrazinamide [23–25]. Three cohort studies of a 6-month regimen of rifampin and isoniazid (without pyrazinamide) for pleural TB found excellent efficacy [26–28], and one study found decreased toxicity, compared to patients treated with these drugs plus pyrazinamide [28]. Studies of the impact of pyrazinamide duration were consistent in finding increased toxicity with treatment for more than 2 months [21,33,34].

There has been great interest in the association between hepatoxicity and the activity of the enzyme primarily responsible for the metabolism of isoniazid, N-acetyltransferase 2 (NAT-2). One clinical trial randomized participants with pulmonary tuberculosis to customized dosing of isoniazid, based on NAT-2 activity (lower dose for slow acetylators, standard dose for intermediate acetylators, higher dose for rapid acetylators) vs. the standard dose and found fewer transaminase elevations with customized isoniazid dosing [35]. However, the definition used for isoniazid-related liver injury (ALT > 2 times the upper limit of normal) was less stringent than commonly used criteria for drug-induced liver injury [15]. Regarding possible replacements for isoniazid, a systematic review found that substitution of fluoroquinolones for isoniazid did not improve safety [36].

The focus of recent clinical trials regarding rifamycins have been the evaluation of higher doses of rifampin and the replacement of rifampin with rifapentine [15]. We did not include studies of higher dose rifampin in this scoping review because there is no indication that increasing the dose of rifampin improves safety and tolerability. In systematic reviews, there was no evidence that using rifapentine [38] or rifabutin [37], rather than rifampin, affects adverse event rates.

Clinical trials have evaluated the impact of early discontinuation of ethambutol (prior to 2 months) based on rapid resistance testing and substitutions for ethambutol on safety and tolerability. Early discontinuation had minimal impacts on adverse events [42,43]. A systematic review found that fluoroquinolones have higher adverse event rates than ethambutol [36]. A trial of faropenem as an alternative to ethambutol showed fewer adverse visual events, but no difference in grade 3 or higher adverse events [45].

Tuberculosis treatment guidelines now recommend daily therapy, based on meta-analyses showing better efficacy for daily vs. intermittent therapy [58,59]. A retrospective cohort study did not show a difference in adverse events with daily vs. intermittent therapy [59], nor did a meta-analysis of trials done among children [47]. However, a recent systematic review of studies in India found that daily treatment was associated with a much higher risk of hepatoxicity than thrice-weekly treatment [48]. Furthermore, a trial of split-dosing (two drugs given on one day, alternating with the other two drugs the next day) showed decreased gastrointestinal and overall adverse events with split dosing, compared to standard daily dosing of all four drugs [50].

More frequent dosing has also been evaluated. Twice-daily dosing (two medications in the morning and the other two in the evening) was associated with decreased gastrointestinal adverse events [49]. One trial compared sequential dose escalation over 18 days to the current standard of simultaneous initiation of the four drugs at full doses among patients with meningitis. Sequential dose escalation was associated with significant decreases in hepatotoxicity and inpatient mortality [51].

**Table 3. Summary of studies of changes in the tuberculosis treatment regimen to prevent adverse events.**

| Regimen change to prevent adverse events | Brief summary |
|---|---|
| Pyrazinamide (PZA) | |
| • Inclusion in the regimen | • Increased adverse events with PZA in one [21], but not the other systematic review [20]<br>• No increased hepatotoxicity, but increased mortality with PZA in elderly patients; RCT [22]<br>• No increased toxicity with PZA in elderly patients without chronic liver disease; three cohort studies [23–25]<br>• Good tolerability of 6-months of rifampin (RIF) + isoniazid) INH for pleural tuberculosis; three cohort studies [26,27], including one that compared to regimen with PZA [28]<br>• Increased toxicity with PZA in patients with liver disease; three cohort studies [29–31] |
| • Dose | • No association between PZA dose and adverse events; systematic review [20]<br>• Flat dosing for adults (1000 mg daily): pharmacokinetic and clinical data suggested as an intervention to decrease adverse events; cohort analysis from an RCT [32] |
| • Duration | • >2 months: increased risk of hepatotoxicity; retrospective cohort study [33] and two systematic reviews [21,34] |
| Isoniazid (INH) | |
| • Dose | • INH dosing by NAT-2 enzyme activity (lower dose for slow acetylators, higher dose for fast acetylators): fewer transaminase elevations; RCT [35] |
| • Comparison to alternative drugs | • Substitution with a fluoroquinolone: no difference in adverse events; systematic review [36] |
| Rifamycin | |
| • Choice of rifamycin | • No difference in adverse events between rifampin vs. rifabutin; systematic review [37]<br>• No difference in adverse events between rifampin vs. rifapentine; systematic review [38]<br>• Rifabutin was tolerated by most patients who had adverse events due to rifampin; three cohort studies [39–41] |
| • Dose of rifamycin | • Similar efficacy and decreased adverse events with lower dose rifabutin (150 mg vs. 300 mg daily); systematic review [37] |
| Ethambutol (EMB) | |
| • Earlier discontinuation | • EMB discontinuation prior to 2 months based on rapid resistance testing: comparable adverse events; two RCTs [42,43] |
| • Comparison to alternative drugs | • Substitution with linezolid (2–4 weeks) – similar adverse events [44], faropenam – decreased side effects, but no difference in grade 3 + adverse events [45]; two RCTs<br>• Substitution with a fluoroquinolone: higher adverse event rates; systematic review [36] |
| Dosing frequency | |
| • Intermittent dosing (<daily) | • No decrease in adverse events compared to daily dosing; retrospective cohort study [46] and a systematic review of trials among children [47]<br>• Increase in hepatotoxicity with daily vs. thrice-weekly dosing; systematic review [48] |
| • Twice-daily dosing | • Decrease in gastrointestinal adverse events but not overall adverse events; RCT [49] |
| • Split dosing (daily treatment with 2 medications every other day, alternating with the other 2 medications) | • Decrease in gastrointestinal and all adverse events; RCT [50] |
| Sequential dose escalation at treatment initiation | • Lower rates of hepatotoxicity and inpatient mortality with sequential dose escalation over 18 days vs. simultaneous, full-dose on treatment initiation for meningitis; RCT [51] |
| Duration of continuation-phase therapy (rifampin + isoniazid) | • 2 vs. 4 months of continuation phase therapy: no decrease in adverse events; four RCTs [52–55].<br>• Regimens using a fluoroquinolone and a shorter continuation phase regimen: similar adverse events compared to standard therapy; systematic review [56] |
| Fixed dose combination formulation vs. separate pills | No difference in serious adverse events or in adverse events resulting in treatment discontinuation; systematic review [57] |

RCT – randomized controlled trials

United States tuberculosis treatment guidelines recommend tests of liver function of all patients prior to initiation of TB treatment but repeat testing during treatment only for patients with specific risk factors for hepatoxicity [60]. However, three cohort studies suggested that routine liver tests for all patients during the intensive phase of therapy was associated with lower risks of severe hepatotoxicity [61–63]. Regarding treatment duration, there was no decrease in adverse events associated with shorter continuation-phase regimens of rifampin and isoniazid (2 vs. 4 months) [52–56].

**Prevention of specific adverse events by interventions other than changes in the treatment regimen**

Though gastrointestinal adverse effects are common during treatment, we found no papers on the effect of commonly-recommended non-pharmacologic measures (dosing at bedtime, dosing with food) [64] and only three clinical trials of pharmacologic interventions (Table 4). Small trials found that a ginger product [65] and an extract from an insect [66] were associated with decreased nausea. A larger trial of two doses of a probiotic showed decreased overall gastrointestinal adverse events [67], though not specifically for nausea. Notably, we did not find any studies of nausea prevention using approved anti-emetic medications.

Prevention of hepatotoxicity was a major focus of studies to improve treatment safety (20 papers, 5 of which were systematic reviews including a large number of clinical trials). Cohort studies of antiviral treatment for chronic Hepatitis B during TB treatment showed that such treatment was associated with decreased risks of hepatotoxicity [68–70]. A number of agents that are thought to be hepatoprotective (medications, herbal preparations, antioxidants, probiotics) have been evaluated as means to decrease hepatotoxicity during tuberculosis treatment. Among medications thought to have hepatoprotective effects, bicyclol [71] and N-acetyl cysteine [72,92] were associated with decreased risks of hepatotoxicity. Statins appeared to be protective in a cohort study [77], but not in two phase 2 trials [75,113]. Among herbal products, silymarin, glycyrrhizic acid preparations, and turmeric showed decreases in liver injury in some systematic reviews [78,81,90,91]. However, two large cohort studies did not suggest hepatoprotection from these agents [88,89]. Limitations in these analyses include different definitions of hepatotoxicity and pooling of different agents into the same meta-analysis [91].

Adjunctive immunomodulator therapy as an intervention to improve treatment outcomes, including adverse events, has been the subject of numerous papers. Corticosteroids were effective in preventing immune reconstitution inflammatory reactions among persons with advanced HIV disease starting antiretroviral therapy in one clinical trial [93]. In systematic reviews, corticosteroids during the treatment of pulmonary disease did not decrease adverse events [94], and corticosteroids increased adverse events in pleural disease [95]. In a single randomized trial of nodal tuberculosis, corticosteroid therapy was associated with a decrease in abscess, sinus, or new adenopathy but an increase in gastrointestinal adverse events [96]. Corticosteroids were associated with decreased mortality in a systematic review of treatment of meningeal tuberculosis among persons without HIV [97], but a survival benefit was not seen among patients with HIV co-infection [98]. Aspirin was associated with decreased neurological adverse events among patients with meningeal tuberculosis in two phase 2 clinical trials [99,100]. Other immunomodulators (Vitamin D, interferon gamma, and several non-standardized products) did not decrease adverse events.

Interventions for alcohol use disorder and diabetes mellitus may be a way to improve treatment outcomes, given the frequency of these conditions among persons with tuberculosis and their association with an increased risk of adverse events [114,115]. However, there have been few studies of this approach. Two studies showed no benefit from naloxone [104,105] or structured counseling among patients with alcohol use disorder [104]. A recent cohort study showed no association between optimized treatment for diabetes and adverse events during tuberculosis treatment [106].

Micronutrient supplementation was evaluated in two clinical trials. A combination of vitamins (including pyridoxine [Vitamin B6]) and selenium among patients being treated for HIV-associated tuberculosis was associated with decreased neuropathy [109]; zinc supplementation did not decrease adverse events in a small trial [110].

**Table 4. Summary of studies of interventions to prevent specific adverse events, other than changes in the treatment regimen.**

| Possible intervention to prevent specific adverse events | Brief summary |
|---|---|
| Gastrointestinal adverse effects | |
| • Non-pharmacologic | • No studies of commonly recommended non-pharmacologic measures (dosing at bedtime, dosing with food) |
| • Supplement/medication prophylaxis | • Ginger [65] and an extract from a stingless bee [66]: decrease in nausea; RCTs<br>• *Lactobacillus casei* probiotic: decrease in gastrointestinal adverse events; RCT [67] |
| Hepatotoxicity | |
| • Screening liver tests during intensive phase treatment | • Lower rates of severe hepatotoxicity in patients who had screening liver tests; three cohort studies [61–63] |
| • Treatment of chronic hepatitis B | • Lower rate of hepatoxicity among patients treated with antiviral drugs; three cohort studies [68–70] |
| • Hepatoprotectants | • Bicyclol decreased liver injury (all grades) and interruption of TB treatment; RCT [71]<br>• N-acetyl cysteine:<br>   ◦ Decrease in hepatotoxicity (all grades) but no effect on severity; systematic review [72]<br>   ◦ Decrease in laboratory markers but not hepatotoxicity; two small RCTs [73,74]<br>• Statins:<br>   ◦ No difference in hepatotoxicity; two RCTs; [75,76]<br>   ◦ Decrease in hepatotoxicity, including severe hepatotoxicity; cohort study [77]<br>• Silymarin:<br>   ◦ Decrease in hepatotoxicity at week 4 of treatment (but not weeks 2 and 8); systematic review [78]<br>   ◦ No significant decrease in hepatotoxicity; cohort study [79] and a small RCT [80]<br>• Glycyrrhizic acid preparations: decrease in hepatotoxicity but impact on severe hepatotoxicity not provided; systematic review [81]<br>• *Lactobacillus casei* probiotic: decrease in alkaline phosphatase but no difference in hepatotoxicity; RCT [82]<br>• Combination of carnatine, alpha-lipoic acid, and co-enzyme Q: decrease in mild liver injury; small RCT [83]<br>• Methionine and vitamin B-complex: decrease in adverse events, including hepatotoxicity; RCT [84]<br>• Combination herbal product (stimuliv): decrease in clinical and subclinical hepatotoxicity; RCT [85]<br>• Combination herbal product (milk thistle, dandelion, barberry, turmeric, and artichoke): decrease in hepatotoxicity; small RCT [86]<br>• Herbal product (Jujube syrup): no significant difference in hepatotoxicity; small RCT [87]<br>• Evaluations of more than one hepatoprotectant<br>   ◦ "Liver care tablet", glucuronolactone, silymarin, and others: no impact on hepatoxicity; cohort study [88]<br>   ◦ Silymarin, glycyrrhetinic acid, and others: no impact on hepatoxicity; cohort study [89]<br>   ◦ Silibinin, carnitine, silymarin, N-acetyl cysteine, garlic: decrease in hepatotoxicity; systematic review [90]<br>   ◦ Silymarin, N-acetyl cysteine, glutathione, and others: decrease in hepatotoxicity, including severe hepatotoxicity; systematic review [91]<br>   ◦ Tumeric + *Tinospora cordifolia*, N-acetyl cysteine, polyherbal product: decrease in hepatotoxicity with turmeric and N-acetyl cysteine [92] |
| Immunomodulatory agents | |
| • Corticosteroids | • Prevention of immune reconstitution inflammatory syndrome (IRIS) with initiation of antiretroviral therapy: decrease in IRIS events and grade 3 events; RCT [93]<br>• Pulmonary: no impact on adverse events; systematic review [94]<br>• Pleural: increase in adverse events; systematic review [95]<br>• Nodal: decrease in abscess, sinus, or new adenopathy but an increase in gastrointestinal adverse events; RCT [96]<br>• Meningitis:<br>   ◦ Decreased mortality, but no effect on other adverse events; systematic review [97]<br>   ◦ Patients with HIV co-infection: no difference in adverse events (and an increase in ALT elevations); RCT [98] |
| • Other immunomodulators | • Aspirin in patients with meningitis: no difference in adverse events, but decreased risk of stroke; two RCTs [99,100]<br>• Vitamin D: no difference in adverse events; two systematic reviews [101,102]<br>• Interferon-gamma: no difference in adverse events; systematic review [29]<br>• Botanical preparations (thought to be immunomodulators): no difference in adverse events [30,31]; two small RCTs<br>• Intravenous quercetin and polyvinylpyrrolidone: decrease in adverse events; cohort study [103] |

*(Continued)*

**Table 4.** (Continued)

| Possible intervention to prevent specific adverse events | Brief summary |
|---|---|
| Other forms of prevention | |
| • Treatment of alcohol use disorder | • Naloxone and behavioral counselling during tuberculosis treatment: no change in adverse events; factorial RCT [104]<br>• Naloxone: no change in adverse events; cohort study [105] |
| • Treatment of diabetes mellitus | • No association between treatment for diabetes and adverse events; cohort study [106] |
| • Metformin | • Adjunctive metformin: no effect on adverse events; two RCTs [107,108] |
| • Micronutrients | • Combination of vitamins (A, B complex, C, E) and selenium: decrease in neuropathy and genital ulcers; RCT [109]<br>• Zinc: no difference in adverse events; RCT [110]<br>• Vitamin E in patients with renal TB: no difference in clinical adverse events; cohort study [111] |
| Acupressure | • Decreased skin reactions; small RCT [112] |

RCT – randomized controlled trial

## Treatment of adverse events

We did not find any studies on the treatment of gastrointestinal adverse events during tuberculosis treatment, neither non-pharmacologic measures (e.g., dosing with food) nor pharmacologic treatment with approved anti-emetic medications (Table 5).

Patients with drug-induced liver injury who have clinical and laboratory resolution after discontinuation of drugs known to be associated with hepatotoxicity (pyrazinamide, isoniazid, rifampin) are often managed with careful re-introduction (re-challenge) of these medications. A systematic review of re-introduction following recovery from hepatoxicity showed trends toward better outcomes among patients who had incremental re-introduction (dose escalation of individual drugs) or sequential full-dose re-introduction, compared with simultaneous full-dose re-introduction [116]. There was no apparent difference based on whether rifampin or isoniazid was the first drug to be re-introduced [116]. Two subsequent cohort studies have shown similar results [117,118]. A third cohort study suggested caution about re-introducing pyrazinamide among patients who have tolerated re-introduction with rifampin and isoniazid [119]. Treatment of chronic Hepatis C allowed patients to resume first-line drugs for tuberculosis in one small cohort study [121]

Drugs and herbal products thought to have hepatoprotective effects have also been evaluated in the treatment of patients who developed treatment-related hepatotoxicity. N-acetyl cysteine may have an impact in reducing the need for liver transplantation or decreasing hospital stay, but there is still uncertainty about the role of this drug [72,143]. Glycyrrhizic acid preparations reduced liver injury in one systematic review, but it is not clear that they decrease clinical events related to hepatotoxicity [81]. Other herbal products evaluated, such as silymarin [122], have not improved clinical outcomes of hepatotoxicity [123].

Immune reconstitution inflammatory syndrome (IRIS) events (paradoxical reactions) are common during the treatment of some forms of tuberculosis. A randomized trial of prednisone for IRIS reactions following antiretroviral therapy initiation was shown to decrease a combined endpoint of hospital days and outpatient therapeutic procedures [128]. Corticosteroids are also commonly used as treatment for other types of immune hyper-reactivity, though with relatively little published data.

Hypersensitivity reactions, often cutaneous, are relatively common and may result in treatment discontinuation in patients with more severe clinical manifestations. Similar to drug-related hepatotoxicity, the primary questions in the management of hypersensitivity adverse events are when and how to safely re-introduce first -line drugs for

**Table 5. Summary of papers on treatment of adverse events.**

| Possible intervention to treat adverse events | Brief summary |
|---|---|
| • Gastrointestinal adverse events | • No studies of commonly recommended non-pharmacologic measures (dosing at bedtime, dosing with food)<br>• No studies on pharmacologic treatment |
| • Hepatotoxicity | • Re-introduction of first-line drugs<br>  ◦ Trend toward better outcomes with incremental (dose-escalation of individual drugs) and sequential (full-dose) than simultaneous (full-dose) re-introduction; systematic review [116]<br>  ◦ Incremental and sequential (full-dose) re-introduction more successful than simultaneous (full-dose); cohort study [117]<br>  ◦ Similar outcomes with incremental and sequential (full-dose); cohort study [118]<br>  ◦ Poor outcomes associated with pyrazinamide re-introduction; prospective cohort study [119]<br>• N-acetyl cysteine: inconclusive impact, may decrease need for liver transplant; systematic review [72]<br>• Bicyclol: more rapid normalization of ALT, no difference in clinical adverse events; small RCT [120]<br>• Treatment of chronic Hepatitis C: associated with ability to re-introduce tuberculosis therapy; small cohort study [121]<br>• Silymarin: no improvement in laboratory or clinical endpoints; systematic review [122]<br>• Glycyrrhizic acid preparations: increase in response rate (resolution of clinical symptoms and a ≥ 50% decrease or normalization of the liver tests), impact on tuberculosis treatment not provided; systematic review [81]<br>• Dihydroxylated bile acid, intravenous SNMC [glycyrrhizin, glycine and L-cysteine], or oral glycyrrhizin: no difference in laboratory of clinical outcomes of hepatotoxicity; cohort study [123]<br>• Herbal product [124], proteolytic enzymes [125], vitamins E and C [126], deoxyribonucleotidum [127]: some improvement in laboratory endpoints but no difference in clinical outcomes; two cohort studies and 2 small RCTs |
| • Immune hyper-reactivity | • Prednisone for treatment of immune reconstitution inflammatory syndrome [IRIS]) after initiation of antiretroviral therapy: decrease in the combined endpoint of days hospitalized and outpatient therapeutic procedures; RCT [128] |
| • Hypersensitivity reactions | • Laboratory tests to identify the causative agent: poor correlation between in vitro lymphocyte reactivity and the results of re-introduction; two cohort studies [129,130]<br>• Safety of patch testing to identify the causative agent among patients with HIV co-infection: most had a systemic reaction (2 were severe); cohort study [131]<br>• Rifabutin for patients who had adverse reactions to rifampin: tolerated by most patients; two cohort studies [39,40]<br>• Protocols for re-introduction following skin rash: adverse reactions during re-introduction were common but seldom serious, most patients tolerated re-introduction of at least some first-line TB drugs; five cohort studies [132–136] and a systematic review [137]<br>• Safety of rechallenge for patients with DRESS syndrome (Drug-Related Drug Reaction with Eosinophilia and Systemic Symptoms): adverse reactions during re-introduction were common, but most patients tolerated re-introduction; three cohort studies [41,138,139]<br>• Safety of re-introduction following Stevens-Johnson syndrome in patients with HIV co-infection: most patients tolerated desensitization [140] |
| • Neurological adverse events | • Fenazid (analogue of isoniazid) among patients with an adverse event attributed to isoniazid: treatment completion without recurrent neurological event; cohort study [141]<br>• Cortexin (polypeptide bioregulator) among patients with prior neurological events due to TB treatment: more rapid recovery from neurological symptoms, more patients were able to restart isoniazid; cohort study [142] |

RCT – randomized controlled trial

tuberculosis. Currently available diagnostic methods have limited sensitivity and specificity. In vitro tests of lymphocyte reactivity correlated poorly with the results of re-introduction [129,130]. Approaches such as patch testing may even inadvertently trigger systemic reactions among patients with HIV co-infection [131]. Re-introduction protocols, using incremental or sequential re-introduction were generally safe and allowed the majority of patients to successfully re-introduce some of the first-line drugs [132–137], including among patients with more severe hypersensitivity manifestations such as the DRESS syndrome (Drug-Related Drug Reaction with Eosinophilia and Systemic Symptoms) [41,138,139].

## Discussion

Major improvements in the safety and tolerability of tuberculosis treatment are likely to come with the identification of new, safer drugs. However, our scoping review demonstrates that there are promising approaches for improving the safety and tolerability of the current standard treatment regimen.

We present a framework to organize the considerable body of literature on this subject: changes in the treatment regimen, other interventions to prevent specific adverse events, and interventions to treat adverse events. We identified 119 studies spanning this broad framework, though > 40% of the 49 clinical trials had < 50 participants/arm, and hence likely to be under-powered. We identified major gaps in research on adverse events, such as the relative lack of research on ways to prevent and treat gastrointestinal adverse events. Finally, we evaluated the research pipeline for interventions to improve safety and tolerability and found it quite limited (only 8 study proposals that have not resulted in published results, five of which were for a single topic [evaluation of customized isoniazid dosing based on NAT-2 status]).

It is notable that, despite its use for more than 40 years and its inclusion in nearly all recommended treatment regimens for rifampin-susceptible TB [60,144], there are still great uncertainties about the safety and tolerability of pyrazinamide. The experience with the two-month regimen of rifampin plus pyrazinamide for tuberculosis preventive treatment clearly demonstrated that this combination of drugs can cause severe and sometimes fatal hepatotoxicity [145]. However, it continues to be recommended as first-line treatment of all forms drug susceptible disease because of its ability to shorten therapy to 6 months [60,144]. The field clearly needs clinical trials that re-evaluate the safety of pyrazinamide; the treatment-shortening trials that led to its routine inclusion in treatment regimens did not include patients with extrapulmonary or paucibacillary tuberculosis. Furthermore, conditions that increase the risk of hepatoxicity (older age, diabetes mellitus, metabolic dysfunction-associated steatotic liver disease [146]) have become more common since these trials were done. Most of the interest to date in identifying alternate forms of treatment for patients with less extensive disease has been in identifying low-risk forms of pulmonary TB for shorter, pyrazinamide-containing therapy [55,147]. However, we found no evidence that doing so will decrease adverse events, because a high percentage of treatment-related adverse events occur within the first 2 months of treatment [52–55]. Our scoping review highlights the need to consider other ways to de-escalate tuberculosis treatment, such as pyrazinamide-free regimens for paucibacillary forms of disease.

Cohort studies have consistently shown that slow NAT-2 acetylators (who have greater exposure to isoniazid) have increased risk of hepatoxicity [148]. Therefore, there is great interest in customized isoniazid dosing based acetylator status. One relatively small trial has been completed, and it suggests that dosing based on acetylator status may decrease isoniazid's effect on the liver [35]. Furthermore, it appears that an additional 5 such trials are underway. However, the limitations of this approach include: the frequency of primary isoniazid resistance in some parts of the world, the programmatic complexity of customized dosing, and the inability to use this approach in settings that use combined formulations.

Daily therapy is now recommended in treatment guidelines, and earlier analyses did not suggest that intermittent dosing (using higher doses of isoniazid, pyrazinamide, and ethambutol) decreased adverse events [59]. However, the recent meta-analysis from India, showing a much higher rate of hepatotoxicity with daily therapy, compared to thrice-weekly therapy, should lead to a re-evaluation of this important question [48]. Our scoping review identified clinical trials of two interventions that may be useful in selected patients who are having difficulty tolerating full-dose simultaneous dosing: split dosing [50] and twice-daily dosing [49].

An approach that deserves additional evaluation is dose escalation over the first 1–2 weeks of therapy, rather than the standard approach of simultaneous full-dose initiation. The small clinical trial of this approach demonstrated statistically significant decreases in hepatotoxicity and early mortality among patients with meningitis [51]. An analogy supporting this approach is that most patients who have had initial recovery from drug-related hepatotoxicity tolerate reintroduction of the same drugs when given by dose escalation [116]. Treatment initiation by dose-escalation has been shown to be effective in other forms of antimicrobial therapy. Clinical trials showed that dose-escalation of trimethoprim-sulfamethoxazole for

*Pneumocystis* prophylaxis decreased adverse events [149,150]. Finally, initial dose escalation is a recommended strategy to improve tolerability of multidrug therapy for pulmonary non-tuberculous mycobacterial infections, particularly among elderly patients [151].

There is an unfortunate paucity of data about how to prevent and treat nausea/vomiting, one of the most common adverse events of tuberculosis treatment. Prevention of nausea/vomiting during cancer chemotherapy, radiation therapy, and post-operative management has been the subject of many clinical trials and systematic reviews [152,153]. Highly effective regimens for preventing nausea/vomiting have markedly improved outcomes of cancer treatment. While some anti-emetic drugs used in chemotherapy have unacceptable drug-drug interactions with rifampin (e.g., the Neurokinin 1 receptor antagonists) [154], other potent anti-emetic drugs have acceptable interactions with rifampin (e.g., ondansetron, olanzapine) [155,156]. There is an urgent need for studies among patients being treated for tuberculosis, based on the extensive experience in these other fields of medicine. There is also a need for studies of the commonly-recommended non-pharmacologic measures for managing treatment related nausea [64]: dosing before sleep, dosing with food.

Two of the most common conditions related to adverse events during tuberculosis treatment are diabetes and alcohol use disorder, estimated to be present in 15% [114] and 30% [115], respectively, of patients globally. Though two small initial studies of treatment of alcohol use disorder did not show benefit [104,105], more effective management of this common condition holds the promise of decreasing hepatotoxicity and other adverse events. Similarly, better treatment of diabetes during tuberculosis treatment may also decrease adverse events, but we found only one retrospective cohort study addressing management of this important area [106].

Though they have been the subject of many clinical trials, cohort studies, and systematic reviews, it is challenging to reach clear conclusions about the role of putative hepatoprotectants in tuberculosis treatment. Many of the trials have small sample sizes, clinical trials and systematic reviews have used different definitions of hepatoxicity, different agents have been combined in some systematic reviews, and there have been very different conclusions from clinical trials and large cohort studies. There is a need for larger clinical trials using commercially available agents and standard definitions of hepatotoxicity.

As is unfortunately true for late-phase tuberculosis clinical trials [15], children < 12 years old were seldom evaluated in these studies of the prevention and management of treatment-related adverse events. Children are commonly thought to have very low risk of adverse events with TB treatment, but recent studies have shown that, when dosed appropriately to match the drug exposures in adults, children have rates of adverse events comparable to adults [157]. We did not systematically assess other important subgroups – such as pregnant women, the elderly, and persons with alcohol use disorder – but we suspect that they, too, have not been adequately included in studies to improve the safety and tolerability of TB treatment.

Our scoping review has at least five limitations. We used a broad search strategy, but it is likely that we did not identify all papers on possible interventions to improve the safety and tolerability of tuberculosis treatment. We could not retrieve 20 papers, but the information in the abstracts for these papers suggest that their inclusion would not have materially changed the findings of this scoping review. We used an artificial intelligence tool to translate papers in languages other than English and Spanish. Doing so increased our inclusion of papers from the full-text review, but there could be problems with the translations. We did not review each of the studies that were included in the 24 systematic reviews included in the scoping review. Finally, as in most scoping reviews, we did not formally assess study quality.

## Conclusions

Treatment-related adverse events remain a major barrier to achieving cure at the individual level and tuberculosis control in the population. In the long term, safer and more effective drugs are needed, but our scoping review demonstrates that there are ample opportunities for studies using currently available drugs and other interventions. We urge the TB community to pursue further high-quality clinical research in this area using a both/and strategy – emphasizing the identification

of safer, better-tolerated new drugs, and doing research on ways to improve the safety and tolerability of the treatment regimens we have now.

## Supporting information

**S1 Table. PRISMA checklist for coping reviews.**
(DOCX)

**S2 Text. Details of the search terms used for the PubMed search.**
(DOCX)

**S3 Table. Characteristics of papers that could not be retrieved.**
(DOCX)

**S4 Table. Characteristics of the papers included in the scoping review.**
(DOCX)

**S5 Table. Randomized trials having sample sizes<50 patients per arm.**
(DOCX)

**S6 Table. Inclusion of persons<18 years of age.**
(DOCX)

**S7 Fig. PRISMA flow chart for the search of clinical trials registries.**
(DOCX)

**S8 Table. Entries in clinical trials registries that include an objective to improve safety and tolerability of rifampin-susceptible tuberculosis treatment and whose results do not appear to have been published.**
(DOCX)

## Acknowledgments

None

## Author contributions

**Conceptualization:** William Burman, David Flynn, James Johnston, C. Robert Horsburgh.

**Data curation:** William Burman, David Flynn.

**Formal analysis:** William Burman.

**Investigation:** William Burman, M. Florencia Martins, Pranay Sinha, C. Robert Horsburgh.

**Validation:** William Burman, M. Florencia Martins, Pranay Sinha, C. Robert Horsburgh.

**Writing – original draft:** William Burman.

**Writing – review & editing:** M. Florencia Martins, David Flynn, James Johnston, Pranay Sinha, C. Robert Horsburgh.

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
