## [Decision Letter · Decision Letter 0]

29 Oct 2025

Dear Dr. Burman,

Thank you for submitting your manuscript to PLOS ONE. After careful consideration, we feel that it has merit but does not fully meet PLOS ONE’s publication criteria as it currently stands. Therefore, we invite you to submit a revised version of the manuscript that addresses the points raised during the review process.

**ACADEMIC EDITOR:**

We look forward to receiving your revised manuscript.

Kind regards,

Selvakumar Subbian, Ph.D.

Academic Editor

PLOS ONE

Journal Requirements:

3. Please remove all personal information, ensure that the data shared are in accordance with participant consent, and re-upload a fully anonymized data set.

Additional Editor Comments:

**REPORT from REVIEWER#3:**

This is an interesting paper which addresses one of the series of questions raised by Dr Burman and his colleagues in their recent efforts to focus more attention on issues of safety and tolerability of current regimens for DS TB. This particular manuscript is a review of published trials and studies assessing interventions to mitigate some of the well known safety/tolerability challenges. As such, I think it represents a valuable contribution to the literature on this issue, and I support acceptance by PLOS ONE, following possible modest revision for the reasons noted below.

That said, I have a number of comments, questions and suggestions for the authors to consider in finalizing their submission.

ABSTRACT:

1. The Conclusions should arise from the Results reported. Several statements in the Conclusions paragraph are not supported by data presented in the Findings statements. One or the other should be revised.

INTRODUCTION:

Line 65: events

METHODS:

Line 125: while it is not unexpected that a detailed analysis of study quality is not within the purview of a scoping review, the fact that sample sizes are often quite small raises concern about study quality. Is it worthwhile to pursue findings of inadequately powered studies?

RESULTS:

Line 156, Table 2: Half of the trials and studies had sample sizes below 60-100 per arm. It would be interesting to know if the risk or odds ratios were of substantial magnitude, even if the p values were not significant (given the relatively low power).

Line 183: Sentence beginning with “However…” lacks its independent clause.

Line 189, Table 3: The apparent finding of opposite effects from PZA in different studies is striking. There is substantial literature on the toxicity of PZA, and on the relation of dose to hepatotoxicity.

Line 218: the authors are certainly aware that intermittent therapy has been linked to lower efficacy. One of the authors co-authored an IPDMA demonstrating this some 15 years ago.

Line 220ff: interesting approaches such as those cited here (split dosing, alternate-day dosing) were suggested long ago, but have not yet been pursued. Such approaches may be of particular interest with drugs linked to toxicity, such as PZA.

Line 292: Table 5: the observation that several studies have found incremental reintroduction safer than simultaneous full dose reintroduction is a very useful observation, thank you

DISCUSSION:

Line 352: while the MRC trials that first demonstrated the treatment shortening capacity of PZA, those trials were followed by several other larger trials (BTA2, USPHS-S21) which involved hundreds of patients. PZA was not included in recommended regimens prior to those trials.

Line 367ff: the reservations about NAT-2 guided therapy seem unduly negative. Especially in areas were slow acetylation is common, adaptation of the dose of INH might be helpful. Studies are underway involving real-time testing and dose-adjustment.

Line 394: unfortunately most of the success of cancer chemotherapy has relied upon drugs that are primarily metabolized by the cytochrome p450 system, esp cyp3A4. Such agents are challenging to use in TB treatment due to inductive features of rifampin and rifapentine (and to a much lesser degree, rifabutin)

Line 400ff: while co-morbid conditions that may affect TB treatment outcomes (alcohol addiction, diabetes, tobacco smoking) have been much discussed, there is quite limited evidence of success in managing these issues while on TB therapy. Coordinated management is complicated and demands additional resources. If the authors have suggestions this would be helpful.

Line 418ff and Line 202: higher doses of drugs (with rifamycins, with PZA, or with children) illustrate the challenge that “there is no free lunch,” with larger doses usually implying greater risk for adverse events.

CONCLUSIONS: These are nicely stated. I particularly appreciated the “both/and” suggestion. I suggest that either the authors consider enlarging on this concept, or that an accompanying editorial address the issues in such an approach. Relevant issues include ways to improve safety of the existing drugs (monitoring, assessing toxicity risk factors for specific drugs, varying the rhythms of drug administration, adding measures to mitigate or prevent adverse events, etc.), ways to improve safety of agents in development (eg. use of humanoid platforms for testing), the advisability of the implicit individualization of regimens (which may be needed to address population-specific vulnerabilities to particular AEs), improved surveillance of the nature and severity of adverse events to allow better assessment of risks and benefits, and mechanisms to support such research due to the substantial lack of commercial support.

FIGURES AND TABLES: There is an inherent contradiction in this MS, in that there are multiple tables presenting multiple interventions whose studies are often very underpowered and of uncertain quality. As a result, I wonder if the tables could be made shorter, with the more detailed information in a supplement.

Reviewers' comments:

Reviewer's Responses to Questions

**Comments to the Author**

1. Is the manuscript technically sound, and do the data support the conclusions?

Reviewer #1: Yes

Reviewer #2: Yes

2. Has the statistical analysis been performed appropriately and rigorously?

Reviewer #1: N/A

Reviewer #2: N/A

3. Have the authors made all data underlying the findings in their manuscript fully available?

Reviewer #1: No

Reviewer #2: Yes

4. Is the manuscript presented in an intelligible fashion and written in standard English?

Reviewer #1: No

Reviewer #2: Yes

Reviewer #1: The manuscript “A scoping review of interventions to prevent and treat adverse events during treatment of rifampin-susceptible tuberculosis” is interesting but requires modification to enhance the presentation and add interest to the readers.

1. Table 1 – Is “Other means of preventing adverse events” placed correctly? Is it referring to

• Nausea/vomiting • Hepatotoxicity • Immunomodulatory agents - Please rephrase the section for clarity

2. In the results, under the section “Changes in the TB treatment regimen “, please try to include subsections as classified in Table 2 and 3

3. Table 2 represents the classification of studies into 4 types. Table 3 has included details drug wise excluding unconventional drugs and limited to PZA, INH and FQ. Table 3 can be elaborated across the key findings – Expand the same information to include additional details like sample size, country, year and type of study in addition to findings

4. Please try to segregate information under appropriate tiles for the section “Prevention of adverse events by interventions other than changes in the treatment regimen”

5. Add additional information to table 4 as well – like table 3

6. Discussion is elaborate need to focus on key messages and how to minimize adverse events. The author should elaborate more on the areas which require more intervention to manage adverse events.

7. Conclusion could add some summary of key findings and elaborate on the interventions or methodologies recommended way forward

8. It would be a good idea to add a graphical summary or a diagrammatic representation of the content by flow or block diagram differentiating intervention strategies targeting the bacterial and/or host mechanism.

Reviewer #2: In this review, the authors have reviewed the existing literature on the management of adverse events associated with anti-tuberculosis drug treatment among rifampicin-sensitive tuberculosis patients, with the objective of identifying interventions (either to prevent or manage those adverse events, either by changing the treatment regimen or by other means) and the ongoing research in this area. Indeed, it is a topic of importance and likely to attract interest, particularly from high TB burden countries. However, this scoping review has a few notable limitations in addition to those listed in the article.

There are several adverse events associated with anti-TB treatment, and they may vary among patients (including in severity); thus, interventions of any kind are unlikely to address all or most of the adverse events. Therefore, most of the studies involve targeted interventions.

Although the authors have summarized several interventions from various studies that evaluated the management of adverse events, it is not clear to what extent these studies achieved successful management of such adverse events.

It would have been interesting for further research if the authors had included key findings from the studies (e.g., percentage reduction in nausea or hepatotoxicity among study participants) for particular types of adverse events (Table 4; e.g., gastrointestinal adverse effects, hepatotoxicity, etc.) rather than simple statements. Otherwise, the table is merely a list of existing literature.

Overall, based on the set of questions framed by the authors, this scoping review provides a comprehensive compilation of the literature.

**Do you want your identity to be public for this peer review?** For information about this choice, including consent withdrawal, please see our Privacy Policy

Reviewer #1: **Yes: ** Radha Gopalaswamy

Reviewer #2: No

---

## [Editor Report · Decision Letter 1]

7 Dec 2025

A scoping review of interventions to prevent and treat adverse events during treatment of rifampin-susceptible tuberculosis

PONE-D-25-52218R1

Dear Dr. Burman,

We’re pleased to inform you that your manuscript has been judged scientifically suitable for publication and will be formally accepted for publication once it meets all outstanding technical requirements.

Kind regards,

Selvakumar Subbian, Ph.D.

Academic Editor

PLOS One
---

## [Editor Report · Acceptance letter]

PONE-D-25-52218R1

PLOS One

Dear Dr. Burman,

I'm pleased to inform you that your manuscript has been deemed suitable for publication in PLOS One. Congratulations! Your manuscript is now being handed over to our production team.

Kind regards,

on behalf of

Dr. Selvakumar Subbian

Academic Editor

PLOS One